# Different Pathogenicity and Transmissibility of Goose-Origin H5N6 Avian Influenza Viruses in Chickens

**DOI:** 10.3390/v11070612

**Published:** 2019-07-04

**Authors:** Kun Mei, Yang Guo, Xuhui Zhu, Nannan Qu, Jianni Huang, Zuxian Chen, You Zhang, Bingbing Zhao, Zhuoliang He, Ming Liao, Peirong Jiao

**Affiliations:** College of Veterinary Medicine, South China Agricultural University, Guangzhou 510642, China

**Keywords:** H5N6, avian influenza virus, pathogenicity, transmissibility, chickens

## Abstract

Highly pathogenic avian influenza H5N6 viruses have been circulating in poultry in Asia since 2013 and producing serious diseases in chickens. Here, we analyzed the genetic properties of 10 H5N6 subtypes AIVs from geese in 2015–2016 in Guangdong province. Phylogenic analysis showed that all HA genes of the 10 viruses belonged to clade 2.3.4.4, and their genes including HA, PA, PB1, M, NP, and NS all derived from Mix-like 1 (CH, VN, LS). Their PB2 genes come from Mix-like 2 (CH, VN, JP). The NA genes were classified into a Eurasian lineage. Therefore, the 10 viruses likely originate from the same ancestor and were all recombinant viruses between different genotypes. We selected A/Goose/Guangdong/GS144/2015(H5N6) (GS144) and A/Goose/Guangdong/GS148/2016(H5N6) (GS148) viruses to inoculate 5-week-old chickens intranasally with 10^4^ EID_50_/0.1 mL dose intranasally to assess their pathogenicity and transmissibility. Inoculated chickens showed that the GS144 virus caused systematic infection with a lethality of 100%, but the lethality of GS148 virus was 0%. The two viruses were efficiently transmitted to contact chickens. The lethality of GS144 and GS148 virus in contact with chickens was 87.5% and 0%, respectively, which suggests that the transmissibility of GS144 virus was stronger than GS148 virus in chickens. Thus, different H5N6 viruses from the same waterfowl can show different pathogenicity and transmissibility in chickens. Continued surveillance and characteristic analysis of the H5N6 viruses will help us to keep abreast of evolution and variation in avian influenza viruses in the future.

## 1. Introduction

Avian influenza viruses (AIVs) are influenza virus type A, which are known to cause various birds’ infections. AIVs are classified into 16 HA and 9 NA subtypes based on their surface glycoproteins: hemagglutinin (HA) and neuraminidase (NA). Some AIVs including H5 and H7 are highly pathogenic avian influenza viruses (HPAIVs) and cause serious diseases in poultry. HPAIVs have been causing severe economic losses to poultry producers and are even a potential threat to human health [1,2].

Most influenza A viruses can infect wild waterfowls, and wild waterfowl are the host reservoir (CDC, Centers for Disease Control and Prevention). Domestic waterfowl are intermediaries in the transmission of AIVs between wild waterfowl and other poultry species [3]. AIVs have a high chance to spread from waterfowl to poultry in farms where these species are cohabitated or in live poultry markets (LPMs). In addition, there may be human involvement in these places, which can enhance the risk of transmission of AIVs to humans from birds [4]. Therefore, domestic waterfowl play an important role in the transmission and spread of AIVs.

Since 2003, the H5 avian influenza has erupted in more than 70 countries in Europe, Asia, Africa, and North America. Indeed, the H5 avian influenza is a real and global threat to the poultry industry. In addition, cases of human infection are also routine and have occurred in more than 10 countries. According to HA phylogenetic analysis, H5 HPAIVs are classified into 0–9 clades and some second clades (World Health Organization/World Organization for Animal Health/Food and Agriculture Organization (WHO/OIE/FAO) H5N1 Evolution Working Group, 2014). The H5N1 viruses of clade 1 are predominant in poultry from Southeast Asia including Vietnam, Laos, Cambodia, Thailand, and China until 2005 [5]. In 2004, the viruses of clade 2.3 began to be detected in China [6]. In 2005, outbreaks of H5N1 from clade 2.2 were seen in wild migratory waterfowl [7,8]. Viruses in clade 2.2 subsequently spread westward, and other viruses with a divergent HA gene (clade 2.3.4) emerged in southern China and began to spread in this region [9]. Viruses from clade 7 were first detected in chickens in China in 2005 (Li et al., 2010). The H5N1 viruses including clades 2.2, 2.3.2, 2.3.4, 4, 7, and 9 co-circulated in China during 2005–2006; clades 2.3.2, 2.3.4, and 7 of the viruses co-circulated in domestic birds and waterfowls during 2007–2009 [7]. The H5N1 HPAIVs of clade 2.3.2.1 first appeared in 2008, and the viruses then gradually appeared to be replacing viruses from clade 2.3.4.4. In addition, the H5NX AIVs including H5N1, H5N2, H5N3, H5N6, and H5N8 subtypes have recently been detected in Chinese poultry and wild birds. Therefore, the H5 AIVs have multiple subtypes and genotypes showing genetic diversities in China [1,2].

On 4 May 2014, the first poultry outbreak caused by H5N6 was reported in Sichuan Province of China (OIE). From 2016 to July 2017, 30 outbreaks or isolates of this subtype of virus were reported in mainland China, Hong Kong, and Taiwan. About 70,000 poultry died and about 300,000 poultry were slaughtered. On 6 May 2014, a fatal H5N6 human infection was reported in Sichuan Province, China. The subject was a poultry dealer working directly in LPMs [1,2].

From 2014 to July 2017, 16 human infection cases of H5N6 HPAIV were reported in China, with 6 deaths/16 infections. This triggered public concerns about the threat of the viruses in terms of economic losses and public health [10]. Thus, it is necessary to understand the genetic properties, pathogenicity, and transmissibility of the H5N6 AIVs. Here, we studied the evolution and molecular characterization of 10 H5N6 viruses from geese in Guangdong Provinces in 2015–2016. We then chose two isolates from geese to experimentally infect SPF chickens for determining their pathogenicity and transmissibility.

## 2. Materials and Methods

### 2.1. Virus and Propagation

The following 10 H5N6 viruses were used in this study: A/Goose/Guangdong/GS013/2015(H5N6) (GS013), A/Goose/Guangdong/GS014/2015(H5N6) (GS014), A/Goose/Guangdong/GS017/2015(H5N6) (GS017), A/Goose/Guangdong/GS018/2015(H5N6) (GS018), A/Goose/Guangdong/GS114/2015(H5N6) (GS114), A/Goose/Guangdong/GS116/2015(H5N6)(GS116), A/Goose/Guangdong/GS119/2015(H5N6) (GS119), A/Goose/Guangdong/GS120/2015(H5N6) (GS120), A/Goose/Guangdong/GS144/2015(H5N6) (GS144), and A/Goose/Guangdong/GS148/2016(H5N6) (GS148). They were all from geese in LPMs of Guangdong province during 2015–2016. The viruses were purified and propagated by three rounds of limiting dilution in 9- to 11-day-old specific pathogen free (SPF) embryonic chicken eggs using standard procedures (Yuan et al., 2014). The allantoic fluid from multiple eggs was pooled, clarified by centrifugation, and stored at −80 ℃. We used a pre-cooled isolation media PBS (pH 7.4) of A/Goose/Guangdong/GS144/2015(H5N6) (GS144) and A/Goose/Guangdong/GS148/2016(H5N6) (GS148) with 10^4^ 50% egg infectious dose (EID_50_) per chicken in 0.1 mL to assess their pathogenicity and transmissibility. The doses were confirmed via the method described by Reed and Muench (1938) using the serial titration of eggs. The experiments were performed in biosecurity level-3 enhanced (BSL-3E) facilities in accordance with established protocols. 

### 2.2. Phylogenetic Analysis

All segments of the 10 Viruses were sequenced. Viral RNA was extracted from allantoic fluid supernatant of embryonic chicken eggs infected with the viruses using Trizol LS Reagent (Invitrogen Life Technologies, Inc. Carlsbad, CA, USA). Reverse transcription was conducted with M-MLV Reverse Transcriptase (Invitrogen Life Technologies, Inc). PCR amplification was performed using specific primers, and the products were purified by QIAquick PCR purification kit (QIAGEN, Valencia, CA, USA). The 8 genes were detected by Shanghai Invitrogen Biotechnology Co., Ltd. The DNA sequences were compiled and edited using the SEQMAN program of Lasergene7.1 (DNASTAR, Madison, WI, USA). The phylogenetic analysis of the 10 H5N6 viruses used the distance-based neighbor-joining method with MEGA 5.2 software (Sinauer Associates, Inc., Sunderland, MA, USA). The reliability of the trees was assessed by bootstrap analysis using 1000 replicates. We found that the horizontal distances of these isolates were proportional to their genetic distance. The nucleotide sequences that we obtained are available from GenBank (accession number: GS013, GS014, GS017, GS018, GS114, GS116, GS119, GS120, GS144, GS148: MN128303-10, MN128311-18, MN128635-42, MN128867-74, MN128648-55, MN128656-63, MN128672-79, MN128664-71, MN128680-87, MN128689-96).

### 2.3. Animal Experiment Design

Two H5N6 viruses A/Goose/Guangdong/GS144/2015(H5N6) (GS144) and A/Goose/Guangdong/GS148/2016(H5N6) (GS148) were used to determine the pathogenicity and transmissibility. The 32 five-week-old SPF White Leghorn chickens were equally divided into two groups and housed in two isolator cages. Eight chickens of each group were infected intranasally. With 100 μL of each virus at an EID_50_ of 10^4^ dose: these were named the GS144-inoculated group and the GS148-inoculated group, respectively. The remaining eight chickens served as a contact group and were housed with the inoculated chickens (GS144 contact group and GS148 contact group, respectively). All chickens were observed daily for 14 days for signs of disease and mortality. 

To study the viral replication, three chickens of each inoculated group were euthanized three days post-inoculation (DPI). Their tissues including the lung, kidney, liver, trachea, spleen, and brain were collected aseptically. 1 g of tissue were homogenized and suspended in 1 mL pre-cooled isolation media PBS (pH 7.4) and took 0.1 mL from the tissue homogenate to test the viral EID_50_. To assess their transmissibility, three chickens from each contact group were euthanized at five DPI, and the tissues mentioned above were collected aseptically. Chickens that died during the 14 days of observation were studied similarly. These tissue samples were homogenized with PBS to determine the titers of viruses in the SPF chicken embryo. Oropharyngeal and cloacal swabs were collected from all chickens at 1, 3, 5, 7, 9, 11, and 13 DPI, and suspended with 1 mL PBS for the detection of viruses shedding. All suspensions were serially diluted with PBS and inoculated into 9–11-day-old embryonic eggs incubated at 37 °C for 48 h. HA titers of the samples were performed using 1% chicken red blood cells according to standard methods. All experiments were performed in BSL-3 biosafety facilities at the College of Veterinary Medicine of South China Agricultural University (SCAU) and were conducted under the guidance of SCAU’s Institutional Animal Care (Guangzhou, Guangdong, China).

### 2.4. Ethics Statements

This study was carried out in ABSL-3 facilities in compliance with approved protocols by the biosafety committee of South China Agriculture University. 

All animals in the experiment were handled in accordance with the principles of the Basel Declaration and recommendations of the approved guidelines of the Experimental Animal Administration and Ethics Committee of South China Agriculture University (SCAUABSL2017-013; 13 June, 2016). The protocol (SCAUABSL2017-013) was approved by the experimental Animal Administration and Ethics Committee of the South China Agricultural University.

## 3. Results

### 3.1. Phylogenic Analysis of the 10 H5N6 Viruses

We analyzed the viruses’ gene sequences including other viruses reported previously from the NCBI GenBank as representative strains. We compared their sequences with our 10 viruses to construct phylogenetic trees. Phylogenetic analyses used the nucleotides sequences of the open reading frames (ORFs) of HA (1704 nt), NA (1431 nt), PB2 (2280 nt), PB1 (2274 nt), PA (2151 nt), NP (1497 nt), and of the combined, overlapping ORFs of M1 and M2 (982 nt) and NS1 and NEP (823 nt).

Sequence analysis showed that all eight segments of the 10 viruses had 97.4–100% nucleotide identity. The HA and NA segments of the 10 viruses shared 97.5–99.9% and 96.8–100% nucleotide identities. The PB1, PB2, and PA segments of the 10 viruses shared 97.4–100%, 98.0–99.9%, and 97.4–100% nucleotide identities. The NP, NS, and M segments of the 10 viruses shared 97.8–99.9%, 99.3–100%, and 98.7–100% nucleotide identities.

Phylogenic analysis showed that all HA genes were clade 2.3.4.4. All of the viruses in Mix-like 1 group were isolated from China, Vietnam, and Laos in 2011 to 2014. Therefore, we defined this group as Mix-like 1. Their HA genes were derived from clade Mix-like 1 (CH.VN.LS) (Figure 1). The viruses in this clade were isolated from goose, chicken, and a variety of ducks like Muscovy duck, breeder duck, wild duck, and Baikal teal. The NA genes and other internal genes (except the PB2 gene) of the 10 viruses were derived from Mix-like 1 (CH.VN.LS) (Figure 2, Figure 3, Figure 4, Figure 5, Figure 6 and Figure 7). Viruses from this clade were isolated from duck and swine. Unlike Mix-like 1 group, their PB2 genes derived from clade Mix-like 2 (CH.VN.JP) including viruses in China, Vietnam, and Japan in 2014 to 2016, we defined this group as Mix-like 2 (Figure 8). Their PB2 genes were isolated from animals including the Northern Pintail Tottori, tundra swan, and duck. Therefore, phylogenetic analysis showed that all the 10 viruses in this study were classified into the Eurasian lineage the 10 viruses may originate from the same ancestor and were all recombinant viruses between different genotypes.

### 3.2. Molecular Characterization of the 10 H5N6 Viruses

The amino acid cleavage site in HA of our 10 viruses has multiple basic amino acids (RERRRK/GLF), which are considered to be characteristic of HPAIVs (Kang et al., 2017b). Here, the amino acid residues were Q226 and G228 in HA, which likely indicated their linkage specificity to the 2,3-NeuAcGal linkage and replicated efficiently in embryonated eggs [11]. In these 10 viruses, we found 5 potential N-linked glycosylation sites in HA1 and 1 in HA2. An extra potential N-linked glycosylation site in HA1 was found in GS018 and GS120 viruses, which the other 8 viruses had none. We also found five potential N-linked glycosylation sites in NA (Appendix A). The NA protein at positions 59-70 had a 12-amino-acid deletion in all 10 isolates—this may increase its pathogenicity in mammals [12,13].

None of the 10 viruses harbored the mammalian-specific substitutions E627K and D701N in PB2 [14,15]. The amino acids in the PA gene were T515 and S224 of the 10 viruses; thus, the absence of T515A and 224P mutations indicated they might not enhance the viral transmission and replication in ducks [16] and duck embryo fibroblasts [17]. The amino acids in the 10 viruses were 615K in PA, and this might increase the adaptation of the H5 virus in mammalian hosts [18]. 

The NS1 protein at positions 80-84 had a 5-amino-acid deletion in all 10 isolates showing increased pathogenicity in chickens and mice [19]. Our 10 viruses were 92E in NS1 genes, and these mutations may increase the pathogenicity and the virulence of the H5 virus in mice [20] and pigs [21], respectively. Our 10 viruses all have the same PDZ-binding motif (PBM) sequences ESEV in NS1, which is an important virulence factor in NS1 and may contribute to murine pathogenicity [22].

Our study showed that the 10 viruses had no substitutions at residues L26, A30, S31, and G34 in the trans-membrane domain of M2 protein. Only the 16,116 virus had V27A at this position, which suggested that only this strain is insensitive to adamantane-based antiviral drugs [23].

### 3.3. Pathogenicity of H5N6 Viruses in Chickens Isolated from Geese

To determine the chicken pathogenicity of the two H5N6 viruses isolated from geese, eight chickens of the two groups were inoculated intranasally with 100 μL 10^4^ EID_50_ allantoic fluid of GS144 and GS148 viruses and observed for 14 days, respectively. The remaining eight chickens were housed with those infected chickens as a contact group. All the chickens inoculated intranasally with GS144 virus began to show clinical typical symptoms at two DPI including facial swelling, eyelid edema, hunching, fluffed feathers, depression, decreased sensitivity to stimuli, huddling behavior, dehydration, comb cyanosis, torticollis, and ataxia. While individual birds may exhibit nervous disorders, all chickens had decreased feed and water intake that rapidly progressed to severe listlessness with death at six DPI. However, chickens inoculated with GS148 virus had no clinical symptoms during the 14 days of observation with no deaths. The lethality of GS144 virus in chickens was 100%, and its mean death time was 3.63 days; the lethality of the GS148 virus in chickens was 0%.

To assess the replication of GS144 and GS148 virus in chickens, we euthanized three chickens inoculated intranasally with the two viruses in each group and collected their tissues at three DPI. The collected tissues included the lung, kidney, liver, trachea, spleen, and brain. All tissues were homogenized with PBS and titrated for virus activity. In GS144 inoculated group, the mean viral loads in tissues at three DPI were 8.25 log_10_EID_50_ in the lungs, 8.08 log_10_EID_50_ in the kidneys, 6.83 log_10_EID_50_ in the livers, 7.25 log_10_EID_50_ in the tracheas, and 6.92 log_10_EID_50_ in the spleens and brains. In the GS148 inoculated group, the mean viral loads in tissues on three DPI of infected chickens were 3.83 log_10_EID_50_ in the lungs, 2.83 log_10_EID_50_ in the kidneys, 3.67 log_10_EID_50_ in the livers, 2.92 log_10_EID_50_ in the tracheas, 2.25 log_10_EID_50_ in the spleens and 2.17 log_10_EID_50_ in the brains (Table 1). Thus, the replication ability of GS144 in chickens was much higher than GS148 although the two viruses were detected in all organs of infected chickens (*p* < 0.05).

To evaluate the shedding of the two H5N6 viruses in chickens, oropharyngeal and cloacal swabs were collected from all chickens at 1, 3, 5, 7, 9, 11, and 13 DPI, and suspended with one ml PBS for the detection of viruses shedding. In the GS144 inoculated group, the GS144 virus was shed from the oropharynx at 1, 3, and 5 DPI, with virus loads were from 2.38 log_10_EID_50_ to 5.07 log_10_EID_50_. It could also be shed from the cloaca at 3 and 5 DPI with virus loads of 2.75 log_10_EID_50_ to 4.03 log_10_EID_50_; all inoculated chickens died within 6 days. In GS148 inoculated group, the virus was shed from the oropharynx at 5, 7, and 11 DPI; the virus loads were 1.75 log_10_EID_50_ to 2.50 log_10_EID_50_. It could also be shed from the cloaca at 11 DPI; the virus loads were 1.75 log_10_EID_50_, and none of the inoculated chickens died. The two viruses shedding titers are shown in Table 2 for each day. Thus, the shedding ability of GS144 in chickens was much stronger than the GS148 virus, although the two viruses could both be shed from the digestive tract and respiratory tract and could be detected in all organs of infected chickens.

### 3.4. Transmissibility of Two H5N6 Viruses in Chickens Isolated from Geese

To determine the transmissibility of the two H5N6 viruses in chickens, eight chickens were housed with the ones inoculated with GS144 and GS148 virus as contact groups in each cage, respectively. These were observed for 14 days. In GS144 group, seven of the eight non-contact chickens housed with the inoculated ones began to show clinical symptoms at four DPI and died in eight days (mortality 87.5%, mean death time 5.5 days). In GS148 group, none of the eight chickens housed with the inoculated ones had clinical symptoms, and none of them died during the 14 days of observation. In the GS144 contact group, the mortality of the chickens was 87.5%, and the mean death time was 5.5 days. There were no deaths in the GS148 contact group. 

To assess the replication of GS144 and GS148 virus in the naive contact chickens, we euthanized three chickens in each contact group to collect tissues included the lung, kidney, liver, trachea, spleen, and brain at five DPI. The results showed that the mean virus loads in the kidney, liver, trachea, spleen, and brain were 8.42, 8.17, 8.17, 7.92, and 7.17 log_10_EID_50_, respectively. In addition, the virus replicated more highly in the lunge (9.17 log_10_EID_50_). The mean virus loads of GS148 virus was 3.17𠄴1.92 log_10_EID_50_, which less than the mean virus loads of GS144 virus (Table 1). Taken together, the replication ability between GS148 and GS144 virus in the naive contact groups were significantly different (*p* < 0.05).

To evaluate the shedding ability of the two H5N6 viruses in the naive contact chickens, oropharyngeal and cloacal swabs were collected from all chickens at 1, 3, 5, 7, 9, 11, and 13 DPI and suspended with one ml PBS for the detection of viruses shedding. In the GS144 naive contact group, the virus was shed from the oropharynx at 3, 5, and 7 DPI, and the virus loads were from 3.75 log_10_EID_50_ to 4.75 log_10_EID_50_. It could also be shed from the cloaca at 5 and 7 DPI with virus loads of 3.63 log_10_EID_50_ to 3.70 log_10_EID_50_; seven-eighths chickens died within eight days. 

In the GS148 naive contact group, the virus was shed from the oropharynx at 3, 7, 9, 11, and 13 DPI with virus loads of 1.75 log_10_EID_50_ to 2.50 log_10_EID_50_. It could also be shed from the cloaca at seven DPI, the virus load was 1.75 log_10_EID_50_, and none of the chickens died. The shedding titers of the two viruses for each day was shown in Table 2. Thus, the shedding ability of GS144 in chickens was much stronger than GS148 virus, although the two viruses were both shed from the digestive tract and respiratory tract and could be detected in all organs of infected chickens. The two viruses both were transmissible, but the transmissibility of GS144 was stronger than GS148.

## 4. Discussion

The first report of a severe outbreak of H5N1 HPAI with 40% mortality [24] occurred on a goose farm in Guangdong of China in 1996. Despite great endeavors to control H5 viruses in China using vaccines, the viruses still circulate and lead to outbreaks in poultry, which contribute to the emergence of various genotypes or sub-lineages. Since 2013, H5 HPAIVs of a new clade 2.3.4.4 with different NA subtypes including H5N2, H5N5, H5N6, and H5N8 strains have been found in domestic and wild birds in China—the prevalence of H5 AIVs is increasing [25]. 

In March 2014, AI in poultry in Laos identified an emergent AIV (H5N6). It was a H5N6 virus derived from assortment with the HA gene of clade 2.3.4.4 H5N1 virus and the NA gene from H6N6 virus. In 13 April 2014 and 23 April 2014, the H5N6 viruses caused several outbreaks in chickens of Vietnam and China, respectively. This led to the death and culling of over 97,000 birds (OIE, 2014). The polybasic HA cleavage sequence PLRERRRKR/GLF of the virus was similar to clade 2.3.4 HPAI viruses. These indicated that the virus was capable of causing high death rates in poultry [26]. The disease has caused human infections in several provinces of China since 2014. These persons have had a history of working directly in LPMs and have had contact with dead poultry before becoming infected [2].

This work offers a phylogenetic tree analysis of eight gene fragments of the 10 H5N6 subtype AIVs from poultry of LPMs in Guangdong Province of China from 2015 to 2016. The analysis suggests that the HA genes of the 10 viruses all belong to clade 2.3.4.4, and their HA, PA, PB1, M, NP, and NS genes all derived from Mix-like 1 (CH.VN.LS). Their PB2 genes came from Mix-like 2 (CH.VN.JP). Their NA genes derived from a Eurasian lineage. Therefore, all 10 viruses may originate from the same ancestor although they came from different hosts and LMPs and were all recombinant viruses between different genotypes. The viruses had similar characteristics likely because of the transport or trade of poultry between different provinces in southern China. 

The multiple basic amino acids RERRRK/GLF of cleavage site and Q226 and G228 in HA showed that our 10 viruses were all HPAIVs [27] and were bound prior to the 2,3-NeuAcGal linkages of AIVs receptors [28]. Different N-linked glycosylation sites in HA show different pathogenicity. We found potential N-linked glycosylation sites at 26, 27, 39, 181, 302, and 500 in HA as well as 190, 135, 59, and 391 in the NA of all 10 viruses. An extra potential site 252 in HA was found in GS018 and GS120 viruses; the other eight viruses did not have this site. 

Several studies have illustrated the relationship between pathogenicity and genes mutations of the viruses. Several amino acid deletions or mutations in the protein PA, PB2, and NS1 may have an important effect on the virulence and adaptation of H5N6 virus in the different hosts. Some deletions found in our 10 viruses such as positions 59–70 (12-amino-acid) in NA and positions 80–84 (5-amino-acid) in NS1—as well as some mutations like 615K in PA and 92E in NS1—might increase the pathogenicity of these viruses in mammals [12]. The PDZ-binding motif (PBM) sequences ESEV found in all 10 viruses might also contribute to pathogenicity in mice [22]. 

Amantadine and rimantadine are adamantane-based drugs used as first-choice antiviral drugs against AIVs. Amantadine-resistant AIVs carried amino acid mutations at residues L26F, V27A/T, A30T/V, S31N/R, and G34E in the trans-membrane domain of M2. Sequence analysis of our other 9 viruses did not show any mutations on these residues despite the GS116 virus having a V27A mutation, which may not be sensitive to adamantane-based drugs only. 

The previous AIV H5N1 virus (A/Hong Kong/156/97) had high (100%) mortality in inoculated chickens and showed a higher propensity to infect and cause disease in geese than in ducks, which was similar to the A/turkey/England/91 (H5N1) virus [29]. That virus also caused 87.5% to 100% mortality in inoculated White Plymouth Rock and White Leghorn chickens in another experiment [30]. From 1999 to 2002, the H5N1 viruses isolated from ducks looked healthy in southern China were HPAIV in chickens, and some of them replicated and were shed in inoculated ducks, but none caused disease signals or death [31]. Four H5N1 AIVs from ducks in 2003 and 2005 in China have been highly pathogenic in SPF chickens. Some of these isolates had low pathogenicity while others were HPAIVs and caused severe infection in mallard ducks [32]. The clade 2.3.4.4 H5N2, H5N6, and H5N8 influenza viruses caused severe infections in ducks [33]. 

The goal of this work was to determine the pathogenicity and transmissibility of these isolates isolated from different hosts in the LMPs of Guangdong in 2016. We selected two geese-derived viruses from the 10 H5N6 viruses that had full genome sequencing, genetic evolution, and molecular characterization. We also verified whether the H5N6 viruses from waterfowls were all HP and could transmit in chickens. To measure pathogenicity of the two H5N6 viruses in chicken, all chickens in infected groups were inoculated with GS144 and GS148 virus at a 10^4 EID_50_ dose, respectively. The mortality rate of GS144 in the inoculated chickens were 100%, while the mortality rate of GS148 in the inoculated chickens were 0%. According to virus loads, we also found that the replication ability between GS144 virus and GS148 viruses in inoculated chickens were significantly different (*p* < 0.05). Thus, the GS144 virus is highly pathogenic and can replicate systemically in the inoculated chickens, which is similar to other experiments with H5N6 viruses in Asia.

However, the lethality of the GS148 virus in chickens was 0%. These two viruses were shed from both the digestive tract and the respiratory tract, but the shedding ability of the GS144 virus was stronger than the GS148 virus in the inoculated chickens. The chickens infected with GS148 virus looked healthy and can expel viruses to the surroundings. This is consistent with the previous conclusion that waterfowl infected with AIVs may remain asymptomatic despite infection and viral shedding. Overall, waterfowl infected with AIVs may remain asymptomatic despite infection and shedding of virions [34].

AIVs could be transmitted via the respiratory system of birds from aerosol and horizontal transmission. In recent years, the predominant AIVs were H5, H7, and H9 subtypes of China. The H9 subtypes have a powerful capacity to transmit horizontally. Only some H5 subtypes could transmit horizontally (World Health Organization, 2013a). In our previous study, three H5N1 viruses A/Goose/Guangdong/1/1996 of clades 0, A/Duck/Guangdong/E35/2012 of clade 2.3.2.1, and A/Chicken/Henan/B30/2012 of clade 7.2 could transmit to the naive contact chickens, but not to ducks or geese from the inoculated group [35]. Some H5N2, H5N6 and H5N8 viruses of clade 2.3.4.4 confirmed from apparently healthy ducks and geese in 2010 to 2014 exhibited efficient direct transmission in ducks (Sun et al., 2016b). In this study, GS144 and GS148 were both sheds from the digestive tract and respiratory tract in contact chickens, but the lethality of chickens in GS144 and GS148 naive contact group was 87.5% and 0%, respectively. The transmissibility of GS144 virus was stronger than GS148 virus in chickens and we sequenced all segments of the two viruses. The results showed that the GS144 and GS148 viruses differ by 26 amino acids (Appendix A). We also compared the signature amino acid mutations of the two viruses and found eight amino-acid changes (Appendix A), which possibly led to the differences between pathogenicity and transmissibility of the virus in chickens [18,36,37,38]. Our results showed that highly pathogenic and mortality in contact chickens. In conclusion, both viruses could transmit horizontally in chickens, and the transmissibility of the GS144 virus was stronger than the GS148 virus.

In summary, this work confirmed that the two H5N6 strains from waterfowl in LMPs in Guangdong province exhibited different pathogenicity and transmissibility in chickens at a 10^4 EID_50_ dose. This result is different from most prior work showing that the H5N6 AIVs was HP in chickens in China. Our study proved that the H5N6 AIVs of clade 2.3.4.4 still circulate in Southern China, and these viruses might be recombinant viruses. H5N6 AIVs from the same hosts may show different pathogenicity and transmissibility in chickens at a certain dose. Our study also laid the foundation for further research on these isolates in the future. Therefore, it is still necessary to study the genetic evolution and mutations of viruses in LPMs in Guangdong province. 

## Figures and Tables

**Figure 1 viruses-11-00612-f001:**
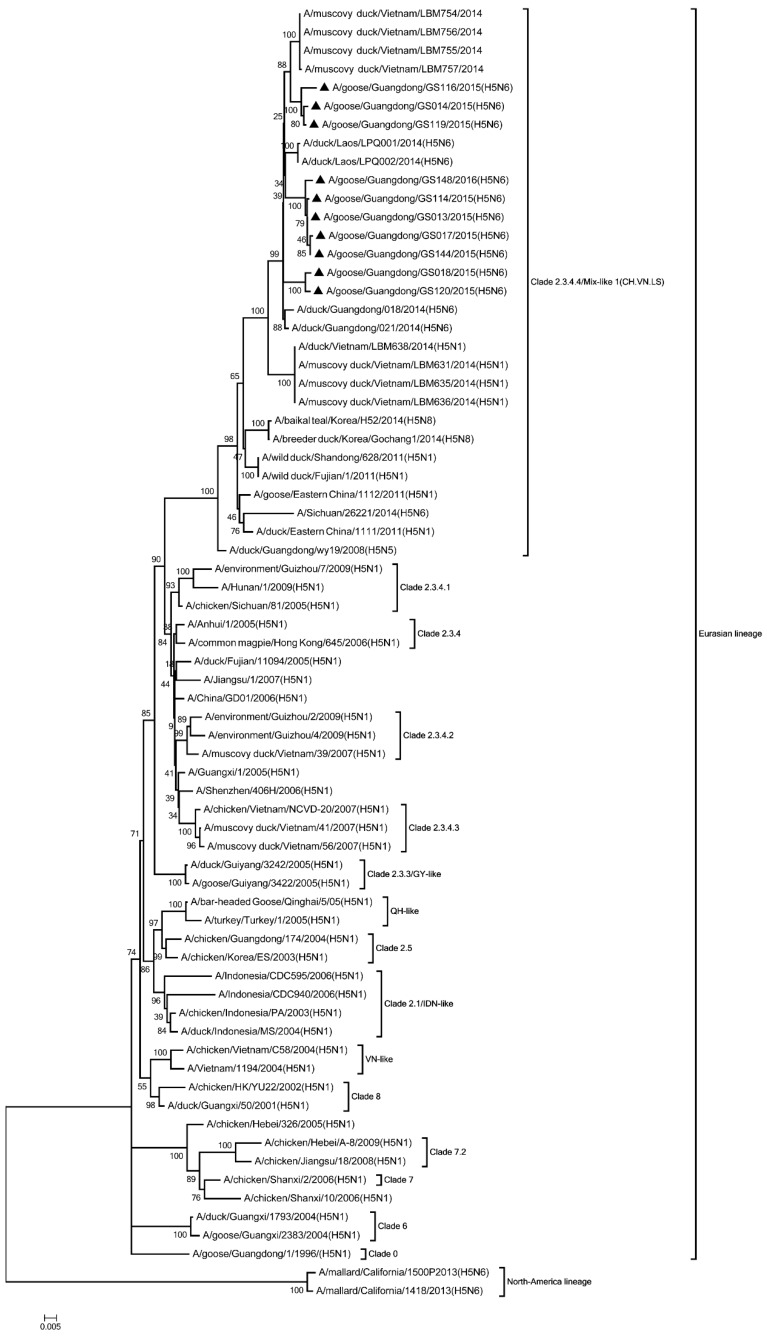
Phylogenetic analysis of HA. Included the completes open reading frame, 1–1704 nucleotides (nt) of HA segment. 10 viruses were characterized in our study belonged to Clade 2.3.4.4/Mix like 1(CH.VN.LS), other virus sequences were downloaded from GenBank. VN, Vietnam; QH, Qinghai; IDN, Indonesia; GY, Guiyang. Our 10 viruses in this study have the black triangle in front of the strains name. The scale bar indicates the branch length and corresponds to 0.005 estimated amino acid substitutions per site.

**Figure 2 viruses-11-00612-f002:**
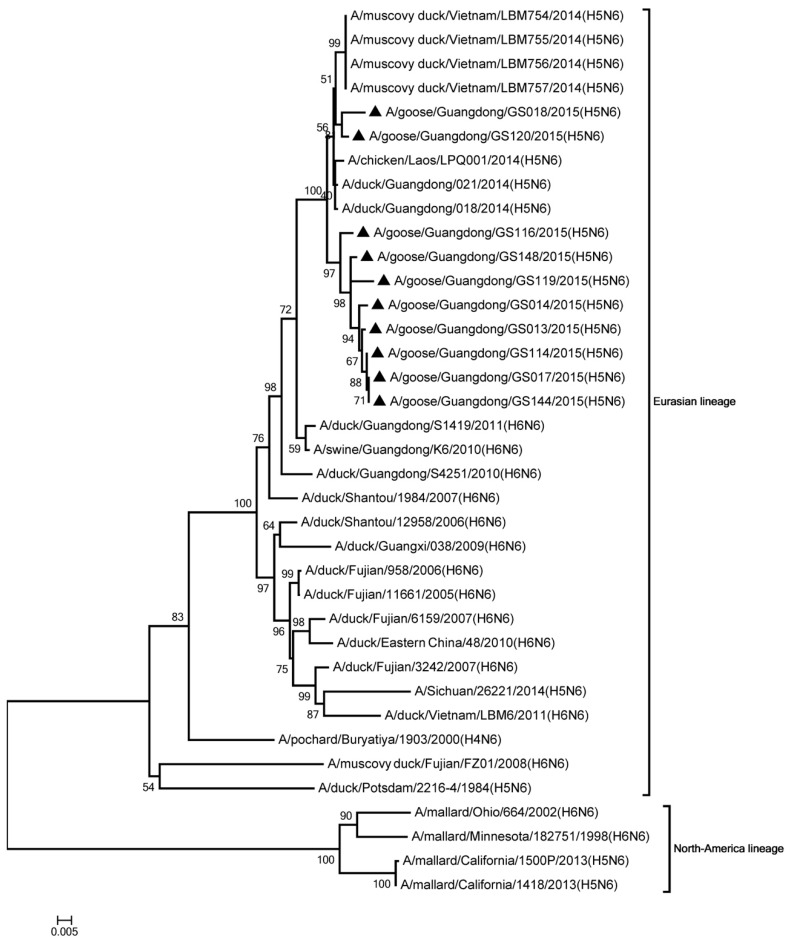
Phylogenetic analysis of NA. Included the completes open reading frame, 1–1431 nucleotides (nt) of NA segment. Except our ten isolates, other virus sequences were downloaded from GenBank. The 10 isolates in this study belonged to Eurasian lineage. Our 10 viruses in this study have the black triangle in front of the strains name. The scale bar indicates the branch length and corresponds to 0.005 estimated amino acid substitutions per site.

**Figure 3 viruses-11-00612-f003:**
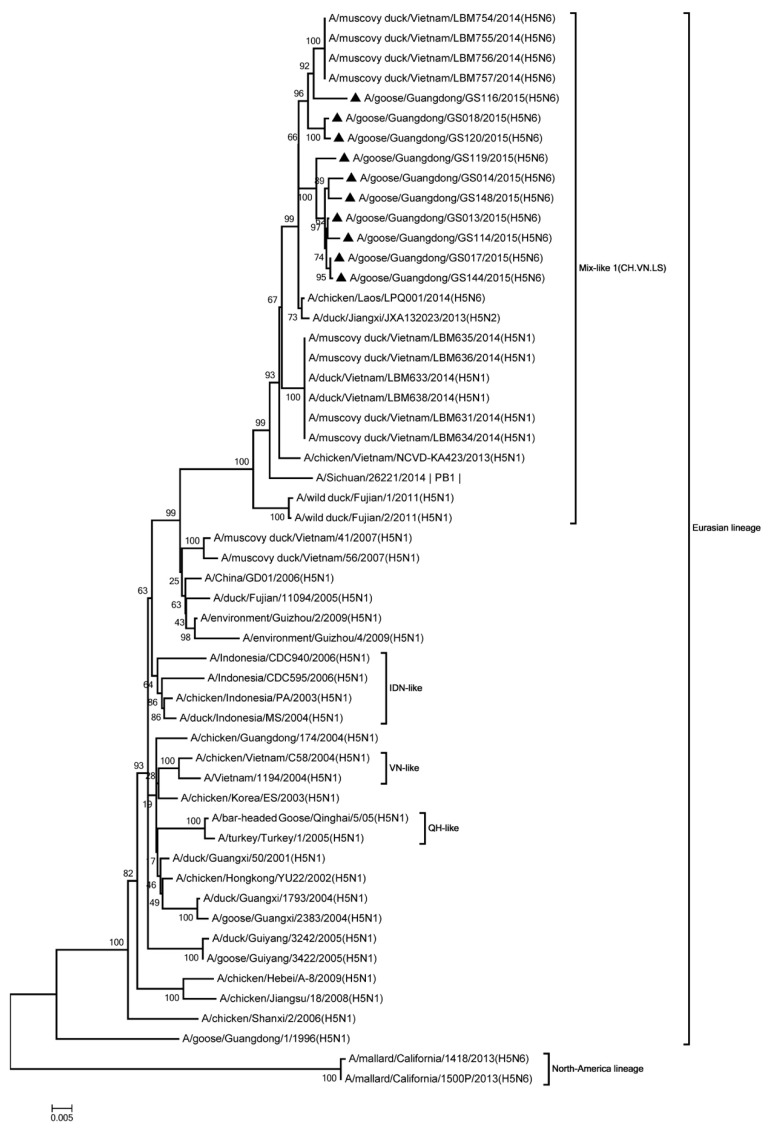
Phylogenetic analysis of PB1. Included the completes open reading frame, 1–2274 nucleotides (nt) of PB1 segment. Except our 10 isolates, other virus sequences were downloaded from GenBank. The 10 viruses characterized in our study belonged to Mix like 1(CH.VN.LS). IDN, Indonesia; VN, Vietnam; QH, Qinghai. Our 10 viruses in this study have the black triangle in front of the strains name. The scale bar indicates the branch length and corresponds to 0.005 estimated amino acid substitutions per site.

**Figure 4 viruses-11-00612-f004:**
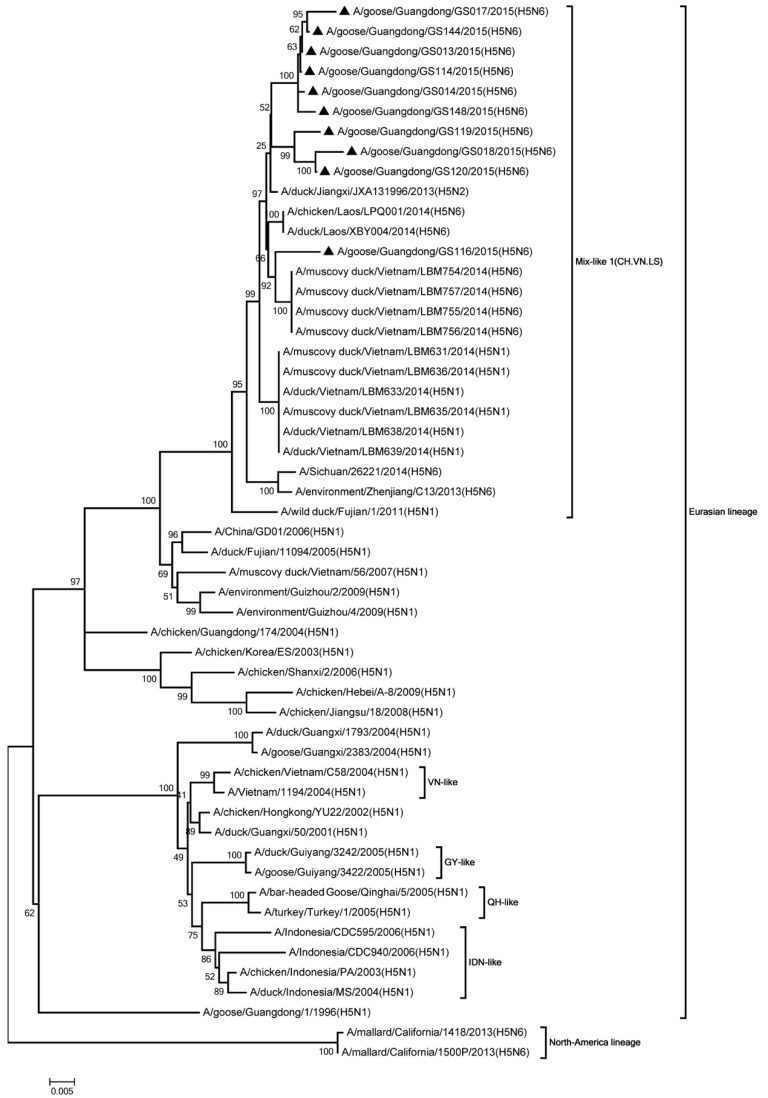
Phylogenetic analysis of PA. Included the completes open reading frame, 1–2151 nucleotides (nt) of PA segment. Except our 10 isolates, other virus sequences were downloaded from GenBank. The 10 viruses characterized in our study belonged to Mix like 1(CH.VN.LS). VN, Vietnam; GY, Guiyang; QH, Qinghai; IDN, Indonesia. Our 10 viruses in this study have the black triangle in front of the strains name. The scale bar indicates the branch length and corresponds to 0.005 estimated amino acid substitutions per site.

**Figure 5 viruses-11-00612-f005:**
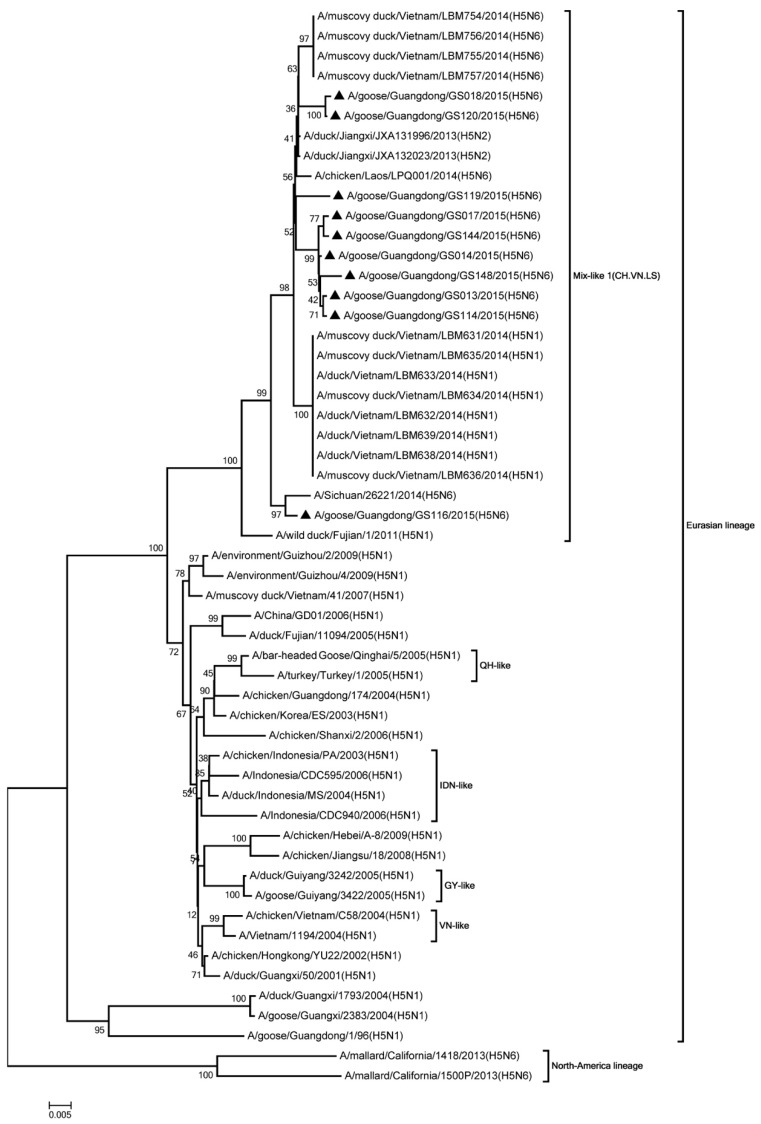
Phylogenetic analysis of NP. Included the completes open reading frame, 1–1497 nucleotides (nt) of NP segment. Except our 10 isolates, other virus sequences were downloaded from GenBank. The 10 viruses characterized in our study belonged to Mix like 1(CH.VN.LS). QH, Qinghai; IDN, Indonesia; GY, Guiyang; VN, Vietnam. Our 10 viruses in this study have the black triangle in front of the strains name. The scale bar indicates the branch length and corresponds to 0.005 estimated amino acid substitutions per site.

**Figure 6 viruses-11-00612-f006:**
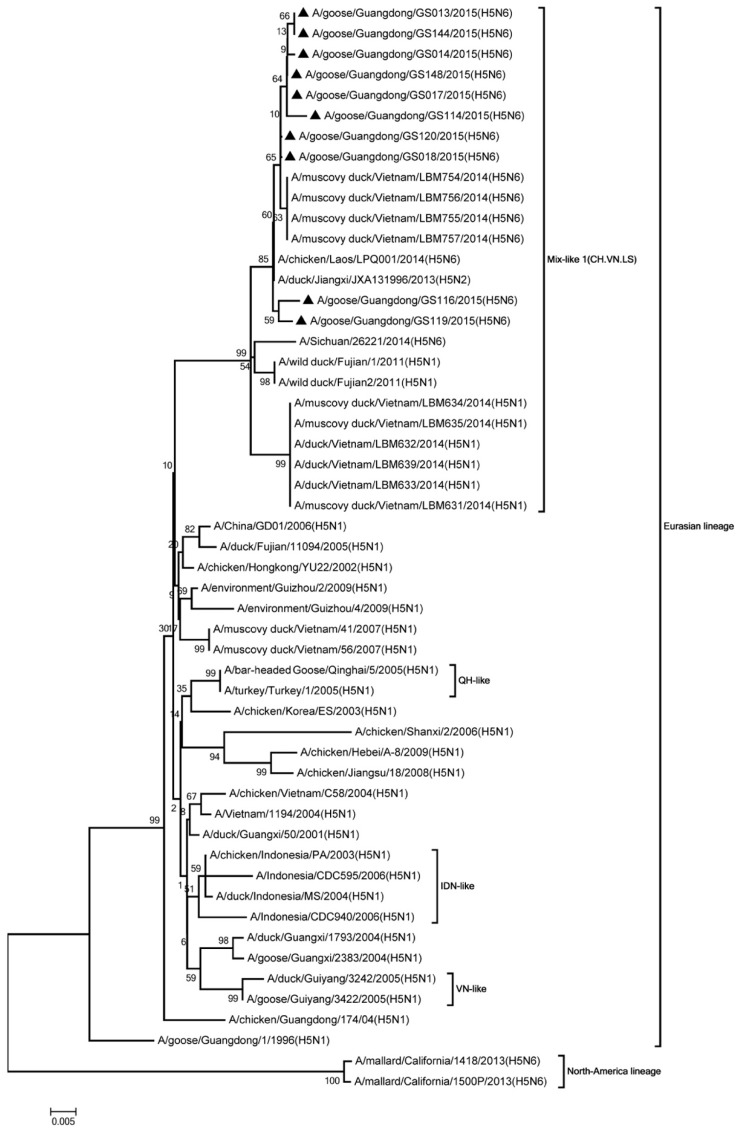
Phylogenetic analysis of M. Concatenated the open reading frames (M1, M2), 1-982 nucleotides(nt) of M segment. Except our 10 isolates, other virus sequences were downloaded from GenBank. The 10 viruses characterized in our study belonged to Mix like 1. QH, Qinghai; IDN, Indonesia; VN, Vietnam. Our 10 viruses in this study have the black triangle in front of the strains name. The scale bar indicates the branch length and corresponds to 0.005 estimated amino acid substitutions per site.

**Figure 7 viruses-11-00612-f007:**
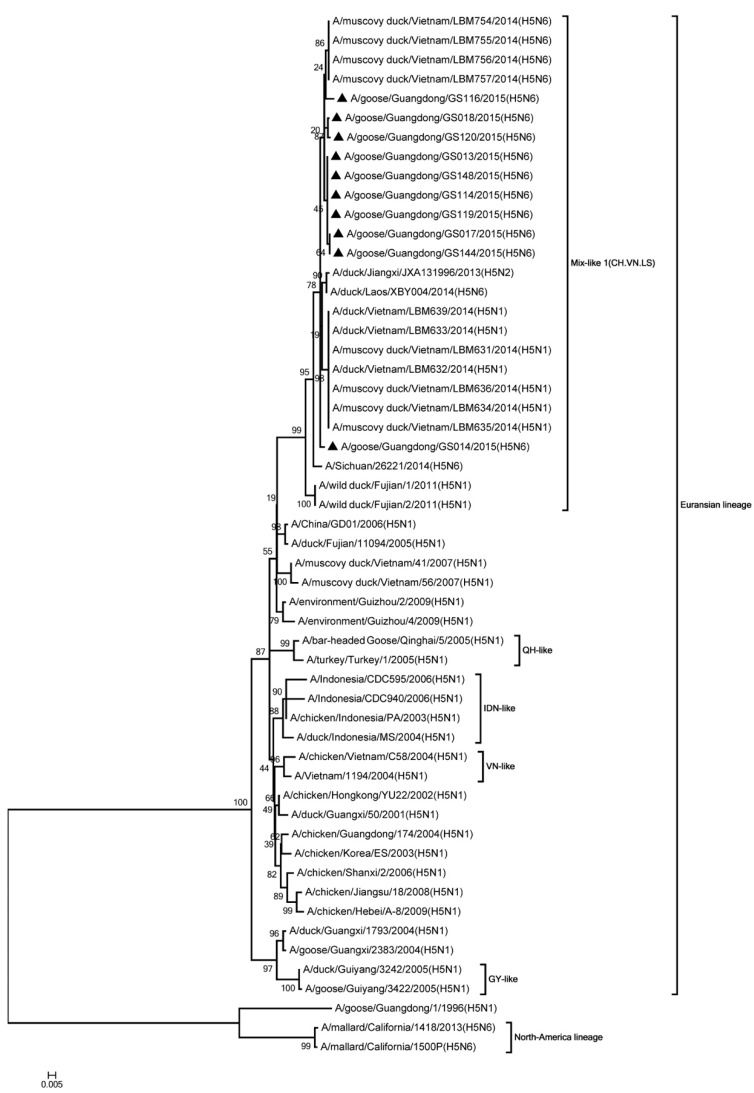
Phylogenetic analysis of NS. Concatenated the open reading frames (NS1, NEP), 1–823 nucleotides (nt) of NS segment. Except our isolates, other virus sequences were downloaded from GenBank. The 10 viruses characterized in our study belonged to Mix like 1(CH.VN.LS). IDN, Indonesia; VN, Vietnam; QH, Qinghai; GY, Guiyang. Our 10 viruses in this study have the black triangle in front of the strains name. The scale bar indicates the branch length and corresponds to 0.005 estimated amino acid substitutions per site.

**Figure 8 viruses-11-00612-f008:**
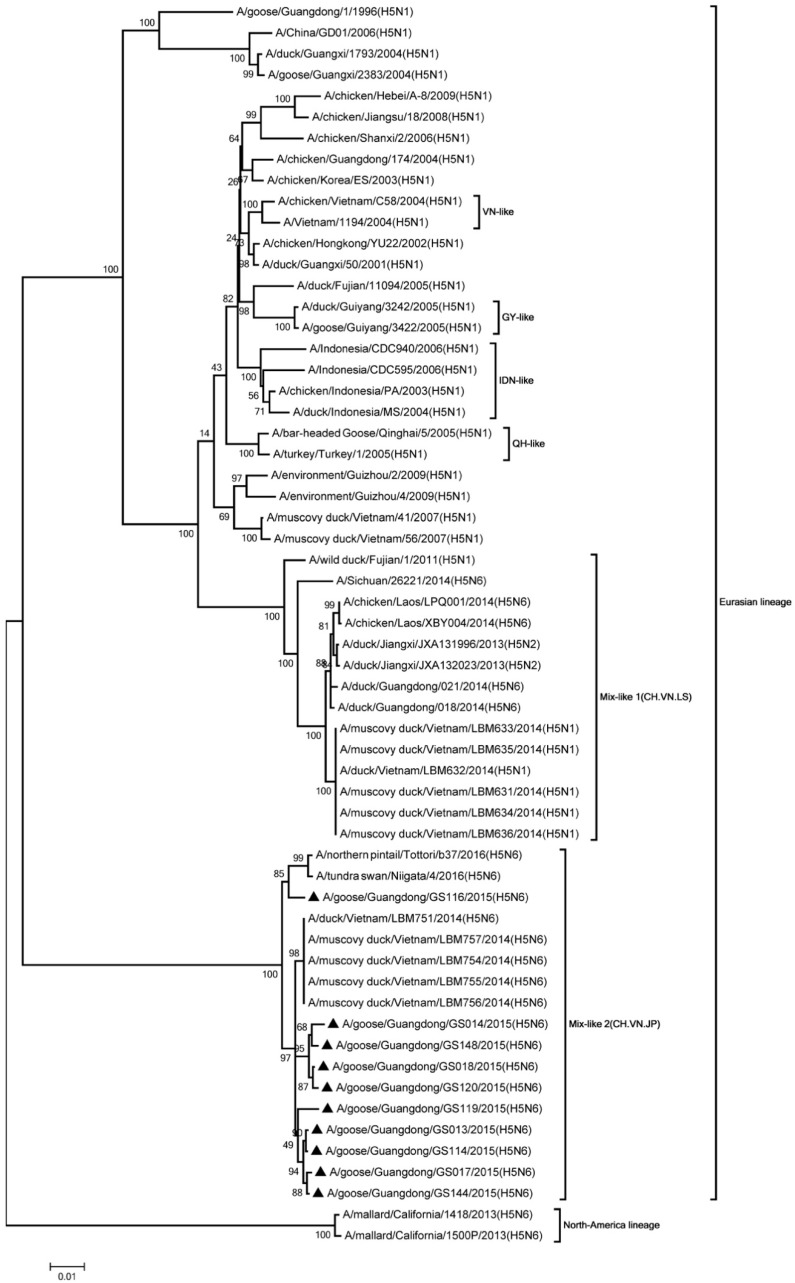
Phylogenetic analysis of PB2. Included the completes open reading frame, 1–2280 nucleotides (nt) of PB2 segment. Except our isolates, other virus sequences were downloaded from GenBank. The 10 viruses characterized in our study belonged to Mix like 2(CH.VN.JP). QH, Qinghai; IDN, Indonesia; GY, Guiyang; VN, Vietnam. Our 10 viruses in this study have the black triangle in front of the strains name. The scale bar indicates the branch length and corresponds to 0.005 estimated amino acid substitutions per site.

**Table 1 viruses-11-00612-t001:** Replication titers in chicken of the two H5N6 viruses after intranasal inoculation.

Strains		Virus Replication Titers in Three SPF Chickens Euthanized at Three DPI and Five DPI (log_10_EID_50_/0.1g) ^b^					
		Trachea	Liver	Spleen	Lung	Kidney	Brain
GS144	Infected ^c^ (3 DPI)	7.25 ± 0.43 ^e^	6.83 ± 0.63 ^e^	6.92 ± 0.52 ^e^	8.25 ± 0.50 ^e^	8.08 ± 1.02 ^e^	6.92 ± 0.52 ^e^
	Contact ^d^ (5 DPI)	8.17 ± 0.58 ^f^	8.17 ± 0.58 ^f^	7.92 ± 0.80 ^f^	9.17 ± 0.38 ^f^	8.42 ± 0.14 ^f^	7.17 ± 0.38 ^f^
GS148	Infected ^c^ (3 DPI)	2.92 ± 1.04	3.67 ± 1.01	2.25 ± 1.09	3.83 ± 0.38	2.83 ± 0.58	2.17 ± 0.38
	Contact ^d^ (5 DPI)	1.92 ± 0.72	3.17 ± 1.52	2.75 ± 1.09	4.17 ± 1.46	3.17 ± 0.58	1.83 ± 0.58

^a^ Five-week-old SPF chickens were inoculated intranasally (i.n.) with 10^4^ EID_50_ of GS144 and GS148 virus in a volume of 100 μL, respectively; three chickens in each inoculated group were euthanized on three DPI and three chickens in each contact group were euthanized on five DPI, the virus loads was determined in samples of trachea, liver, spleen, lung, kidney and brain in eggs. ^b^ For statistical analysis, a value of 1.5 was assigned if the virus was not detected from the undiluted sample in three embryonic hen eggs. Virus loads are expressed as means standard deviation in log_10_EID_50_/0.1g of tissue. ^c^ Chickens inoculated with virus. ^d^ Naive contact chickens housed with those inoculated. ^e^
*p* < 0.05 compared with the titers in the corresponding organs of the GS148-inoculated chickens. ^f^
*p* < 0.05 compared with the titers in the corresponding organs of the GS148-contacted chickens. In these cases, a *p*-value of less than or equal to 0.05 was considered statistically significant.

**Table 2 viruses-11-00612-t002:** Virus loads in oropharyngeal swabs (Oro) and cloacal swabs (Clo) from chickens.

Strains		Days Post-Inoculation (log_10_EID_50_/0.1 mL) ± SD ^a^													
		1 Day		3 Day		5 Day		7 Day		9 Day		11 Day		13 Day	
		Oro	Clo	Oro	Clo	Oro	Clo	Oro	Clo	Oro	Clo	Oro	Clo	Oro	Clo
GS144	Infected ^b^	2.38 ± 0.18 (2/8)	ND ^d^ (0/8)	5.07 ± 0.79 (7/8)	4.03 ± 1.25 (8/8)	3.50 (1/1)	2.75 (1/1)	-	-	-	-	-	-	-	-
	Contact ^c^	ND (0/8)	ND (0/8)	3.75 (1/8)	ND (0/8)	4.05 ± 0.65 (5/8)	3.70 ± 1.41 (5/8)	4.7 ± 0.5 (3/4)	3.63 ± 1.59 (2/4)	ND (0/1)	ND (0/1)	ND (0/1)	ND (0/1)	ND (0/1)	ND (0/1)
GS148	Infected ^b^	ND (0/8)	ND (0/8)	ND (0/8)	ND (0/8)	1.75 (1/8)	ND (8/8)	2.50 (2/8)	ND (0/8)	ND (0/8)	ND (0/8)	2.50 (1/8)	1.75 (1/8)	ND (0/8)	ND (0/8)
	Contact ^c^	ND (0/8)	ND (0/8)	ND (0/8)	ND (0/8)	ND (0/8)	ND (0/8)	1.75 (1/8)	1.75 (1/8)	2.2 (1/8)	ND (0/8)	2.5 ± 1.06 (2/8)	ND (0/8)	1.75 (1/8)	ND (0/8)

^a^ For statistical purposes, a value of 1.5 was assigned if virus was not detected from the undiluted sample in three embryonic hen’s eggs (Sun et al., 2011). ^b^ Chickens inoculated with virus. ^c^ Naive contact chickens housed with those inoculated. ^d^ Not detected.

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
