# Peer review of "Different Pathogenicity and Transmissibility of Goose-Origin H5N6 Avian Influenza Viruses in Chickens"

_viruses, 2019, doi:10.3390/v11070612_

Round 1
Reviewer 1 Report
In this manuscript by Kun Mei et al., the authors analyzed the genome of ten Highly Pathogenic Avian Influenza viruses of subtype H5N6 that were isolated in geese in the Guangdong province of China in 2015 and 2016. Two of the ten isolates were compared as regards their pathogenicity and transmissibility in a chicken model. Strikingly, the two viruses showed very distinct pathogenicities (lethality rates of 87% and 0%), although they were both transmissible to contact chickens.
Main Findings
This manuscript, essentially descriptive, is interesting, but at this stage it is very preliminary and incomplete. There is no in-depth analysis of the striking difference of pathogenicity between the two viruses. The authors have determined the full-length sequence of the ten virus isolates, and therefore they probably have complementary informations that could point to virulence determinants distinguishing GS144 and GS148 (at least candidate substitutions that could explain the striking difference).One interesting information is how easy it seems to find a low-virulence mutant within a limited set of Highly Pathogenic Avian Influenza virus isolates. Do these different levels of pathogenicity reflect the reality (different pathogenicities of the original isolates), or could these be laboratory artefacts resulting from the viruses’ propagation by three rounds of passage in embryonic eggs?
Minor remarks
Line 92. Replace 104 by 10^4.Line 97. The genetic sequences … were .Line 99. … with MLV (Invitrogen…)Lines 144-46. Phylogenetic analyses were performed using 1704 nucleotides (nt) of the HA segment, 1431 nt of NA (neuraminidase), ….etc (nt is more appropriate than bp in this context).Line 201. …G228 in HA, likely indicative of their specificity for avian-type receptors with the 2,3-NeuAcGal linkage…Lines 226-27. ..intranasally with EID50 of either GS144 or GS148 virus and observed for 14 days.Line 254. Virus titers are expressed as standard deviation log10 units of EID50.Lines 240-41. ..the mean viral load at 3 DPI (expressed as log10 of EID50) was 8.25 in the lungs, 8.08 in the kidneys…Lines 283-297. Same remark as above. The average viral load in tissues (expressed as log10 of EID50) was 9.17 in the lungs…Tables 1 and 2. “Contact” (not contacted)Line 273. In the GS144 group, seven of the eight contact chickens….
Author Response
Dear Editor:
Thank you for considering our Manuscript ID Viruses-511752 entitled " Different Pathogenicity and Transmissibility of Goose-origin H5N6 Avian Influenza Viruses in Chickens". We are pleased to resubmit the revised version of our manuscript after considering the reviewers’ comments. We greatly appreciate the effort and time taken by you and the reviewers to evaluate our work. We have addressed all the concerns of the reviewers, and we offer detailed responses to their comments below. All changes are highlighted in yellow in the revised manuscript.
Sincerely,
Peirong Jiao
Ph.D., Professor
College of Veterinary Medicine,
South China Agricultural University,
Guangzhou, 510642, China.
E-mail: prjiao@scau.edu.cn
------------------------------------
Response to Reviewer 1 Comments
Point 1: This manuscript, essentially descriptive, is interesting, but at this stage it is very preliminary and incomplete. There is no in-depth analysis of the striking difference of pathogenicity between the two viruses. The authors have determined the full-length sequence of the ten virus isolates, and therefore they probably have complementary information’s that could point to virulence determinants distinguishing GS144 and GS148 (at least candidate substitutions that could explain the striking difference). One interesting information is how easy it seems to find a low-virulence mutant within a limited set of Highly Pathogenic Avian Influenza virus isolates. Do these different levels of pathogenicity reflect the reality (different pathogenicities of the original isolates), or could these be laboratory artefacts resulting from the viruses’ propagation by three rounds of passage in embryonic eggs?
Response 1: Thank you for your review and your professional suggestion. The 10 isolates of in the study were purified and propagated by three rounds of limiting dilution in the allantoic cavity of 9 –11 days old specific-pathogen-free (SPF) embryonated chicken eggs. Our 10 viruses in this study confirm that no mutations were introduced by sequencing during the passage to future study the molecular characteristics and pathogenicity between those viruses. We added the four Tables (S2-S5) about the amino acids difference between GS144 and GS148 viruses.
In our previous studies, when the chickens were infected by HPAIVs with 106EID50 dose, we found that most of HPAIVs have high pathogenicity and high replication titter in chickens, and there is no obvious difference in replication and pathogenicity. Here, when the chickens were infected by HPAIVs with 104EID50 dose, these viruses may show different pathogenicity and transmissibility in chickens to future research molecular determination of HPAIVs in pathogenicity.
Point 2: Line 126. Embryonic eggs incubated at 37°C for 30 hours. This short incubation time may be too short for some viruses. As a result, some eggs may score negative while a longer incubation time may have revealed a positive status, which could in turn considerably bias the viral titers.
Response 2: Thank you for your review. We are sorry that we make a mistake. We have corrected the descriptions in the line 126-128 of the revised manuscript. When isolated the AIVs, the 9-11-day-old embryonic eggs were inoculated by samples of AIVs and incubated at 37°C for 48 hours. However, some of the incubated embryonic eggs inoculated by samples of HPAIVs died in 30 hours. Therefore, these inoculated embryonic eggs incubated really at 37°C for 30 hours.
Point 3: What exactly is Mix-like 1 (CH.VN.LS)? Is it a lineage?
Response 3: Thank you for your review. According to the reviewer 2 opinions, we added two new viruses A/mallard/California/1418/2013 (H5N6) and A/mallard/California/1500P /2013(H5N6), and they were classified into theNorth American lineage. We also added four newH5N6 subtype viruses A/Muscovy duck/Vietnam/LBM754/2014(H5N6), A/Muscovy duck/Vietnam/LBM755/2014(H5N6), A/Muscovy duck/Vietnam/LBM756/2014(H5N6), A/Muscovy duck/ Vietnam/LBM757/2014 (H5N6), and they and other viruses, which were not belong to the North American lineage, were classified into the Eurasian lineage. Our 10 viruses belong to the Mix-like 1 group, and all of the viruses in Mix-like 1 group were isolated from China, Vietnam, and Laos in 2011 to 2014. So, we define this group as Mix-like 1.
We have added the information in the line 149-151 of the revised manuscript.
Point 4:“…had 97.5% and 99.9% homology.” What exactly is the meaning of 97.5 and 99.9 (speaking of ten viruses)? And is it “homology”, or more probably “nucleotide identity”? More accurately, perhaps the authors should write “the ten HA genes were very similar, any two sequences sharing 97.5 to 99.9% of identical nucleotides.”
Response 4: Thank you for your review. We have corrected the descriptions as your opinions in the line 154-155 of the revised manuscript.
Point 5: Lines 152-166. Same remark as above. “Homology” should probably be replaced by “% of identical nucleotides”.
Response 5: Thank you for your review. We have corrected the descriptions as your opinion in the line 156-169 of the revised manuscript.
Point 6: Line 162. What exactly is Mix-like 2 (CH.VN.JP)? Is it a lineage?
Response 6: Thank you for your review. According to the reviewer 2 opinions, we added two new viruses A/mallard/California/1418/2013 (H5N6) and A/mallard/California/1500P /2013(H5N6), and they were classified into theNorth American lineage. We also added four newH5N6 subtype viruses A/Muscovy duck/Vietnam/LBM754/2014(H5N6), A/Muscovy duck/Vietnam/LBM755/2014(H5N6), A/Muscovy duck/Vietnam/LBM756/2014(H5N6), A/Muscovy duck/ Vietnam/LBM757/2014 (H5N6),and they and other viruses, which were not belong to theNorth American lineage, were classified into the Eurasian lineage. The PB2 genes of our 10 viruses belong to the Mix-like 2 group, and all of the viruses in Mix-like 2 group were isolated from China, Vietnam, and Japan in 2014 to 2016, unlike Mix-like 1 group were isolated from China, Vietnam, and Laos in 2011 to 2014. So, we define this group as Mix-like 2.
We have added the information in the line 165-167 of the revised manuscript.
Point 7: Figures 1-8. (i) What exactly is the meaning of the scale bar (0.005)?
Response 7: Thank you for your review. The phylogenetic tree was constructed using the neighbor-joining (NJ) method within MEGA software (version 5.2). The scale bar indicates the branch length and corresponds to 0.005 estimated amino acid substitutions per site.
Point 8: (ii) Which are the ten viruses from the present study? (presumably, the isolates with a black triangle)
Response 8: Thank you for your review. We are sorry that we missing this information. Our 10 viruses have the black triangle in front of the strains name. We have added the information in the figure 1 to 8 of the revised manuscript.
Point 9:(iii) what are the boundaries of the sequences? (for instance, in the case of HA: the full open reading frame (1704 nucleotides), starting from the A of the first codon of HA).
Response 9: Thank you for your review. The phylogenetic analysis based on each segments ORF of the 10 viruses. We corrected the descriptions in the figure captions as your opinions in figures 1 to 8 of the revised manuscript
Point 10: Lines 203-206. What exactly is the meaning of 26NNS, 27NST, … etc?
Response 10: Thank you for your review. We are terribly sorry that our descriptions make you confused. We have corrected the descriptions as your opinions in the line 223-226 of the revised manuscript. We added the new table to show the summary of the potential glycosylation sites of the GS144 and GS148 viruses in supplementary information Table S2.
The potential glycosylation sites require the context of the amino acid combination (N-X-S/T) to be glycosylated, which can impact protein folding, receptor binding, fusion activity of the virus.
Point 11: Lines 209-210. None of the ten viruses harboured the mammalian-specific substitutions E627K and D701N in PB2.
Response 11: Thank you for your review. We have corrected the content as your opinions in the line 229-230 of the revised manuscript.
Point 12: Line 216-218. According to the authors, Ser42 in NS1 is associated with increased pathogenicity in mice. This is largely exaggerated and in fact inaccurate. Actually position 42 in NS1 is always a Serine (with very rare exceptions), and Jiao et al. (2008, PMID: 18032512) characterized one H5N1 isolate [A/Duck/Guangxi/12/2003, “DK12”] that was considerably attenuated in mice, as a result of substitution S42P in NS1. In short, proline substitution of the strictly conserved residue S42 dramatically reduces the pathogenicity in mice [this only shows the importance of S42, and in no way implies that S42 is associated with increased pathogenicity in mice]
Response 12: Thank you for your review. We are sorry that we make a mistake. We have corrected the descriptions as your opinions in the line 237-239 and line 380-381of the revised manuscript. The substitution S42P in NS1 has reduced the pathogenicity in mice, but the S42 in NS1 was unable to increased pathogenicity in mice.
Point 13: Lines 240-247 and 254-255. What exactly is the meaning of EID50/0.1 ml of tissue? The tissues were homogenized in PBS, and the viral loads should be expressed either as EID50/tissue, or EID50/mg of tissue.
Response 13: Thank you for your review. For the meaning of EID50/0.1 ml: 1 g of tissue were homogenized and suspended in 1 ml pre-cool isolation media PBS (pH 7.4). and took 0.1 mL from the tissue homogenate to test the viral EID50/0.1 ml. We can use either EID50/0.1 ml or EID50/0.1 mg to measure of the viral replication ability.
We have added the information in the line 120-121 of the revised manuscript.
Point 14: The discussion does not provide interesting information, and does not even provide hypotheses to try to explain the difference of pathogenicity between the two viruses. At least the comparison of the two full genomic sequences of GS144 and GS148 could point to candidate substitutions associated with the distinct levels of pathogenicity.
Response 14: Thank you for your review. According to your and reviewer 2's opinion, we added the discussion about pathogenicity differences between GS144 and GS148 virus in the line 404-411 of the revised manuscript. We also added the four new Tables (S2-S5) about the amino acids difference between GS144 and GS148 viruses in the supplementary information. Moreover, we have added the statistical analyses in the table 1 of the revised manuscript.
Point 15: Line 92. Replace 104 by 10^4.
Response 15: Thank you for your review. We have corrected the content as your opinions in the line 92 of the revised manuscript.
Point 16: Line 97. The genetic sequences … were.
Response 16: Thank you for your review. We have corrected the content as your opinions in the line 97 of the revised manuscript.
We added a new Table S1 showed that the 8 segments of our 10 viruses were sequenced in the Supplementary information.
Point 17: Line 99. … with MLV (Invitrogen…)
Response 17: Thank you for your review. We have corrected the descriptions as your opinions in the line 99-100 of the revised manuscript.
Point 18: Lines 144-146. Phylogenetic analyses were performed using 1704 nucleotides (nt) of the HA segment, 1431 nt of NA (neuraminidase), … etc (nt is more appropriate than bp in this context).
Response 18: Thank you for your review. We have corrected the content as your opinions in the line 145-148 of the revised manuscript.
Point 19: Line 201. …G228 in HA, likely indicative of their specificity for avian-type receptors with the 2,3-NeuAcGal linkage…
Response 19: Thank you for your review. We have corrected the content as your opinions in the line 222-223 of the revised manuscript.
Point 20: Lines 226-227. ...intranasally with EID50of either GS144 or GS148 virus and observed for 14 days.
Response 20: Thank you for your review. We have corrected the content as your opinions in the line 246-248 of the revised manuscript.
Point 21: Line 254. Virus titers are expressed as standard deviation log10units of EID50.
Response 21: Thank you for your review. We are sorry about the incomplete descriptions. All related errors have been revised, and the results are highlighted in the manuscript.
Point 22: Lines 240-241. ...the mean viral load at 3 DPI (expressed as log10of EID50) was 8.25 in the lungs, 8.08 in the kidneys…
Response 22: Thank you for your review. We have corrected the content as your opinions in the related errors, and the results are highlighted in the manuscript.
Point 23: Lines 283-297. Same remark as above. The average viral load in tissues (expressed as log10of EID50) was 9.17 in the lungs…Tables 1 and 2. “Contact” (not contacted)
Response 23: Thank you for your review. Commonly, most of the studies define the term as a contact group. Here are the references.
Maines, T. R., Chen, L. M., Matsuoka, Y., Chen, H., Rowe, T., Ortin, J., ... & Tumpey, T. M. (2006). Lack of transmission of H5N1 avian–human reassortant influenza viruses in a ferret model. Proceedings of the National Academy of Sciences, 103(32), 12121-12126.
Gao, Y., Zhang, Y., Shinya, K., Deng, G., Jiang, Y., Li, Z., ... & Liu, L. (2009). Identification of amino acids in HA and PB2 critical for the transmission of H5N1 avian influenza viruses in a mammalian host. PLoS pathogens, 5(12), e1000709.
Are we finally going to make this change?
Point 24: Line 273. In the GS144 group, seven of the eight contact chickens….
Response 24: Thank you for your review. We have corrected the content as your opinions in the line 298 of the revised manuscript.

Reviewer 2 Report
The manuscript by Mei et al., describing different pathogenicity and transmissibility of two H5N6 strains from geese provides some insight into the diverse nature of avian influenza strains. However, there is some detail missing from the manuscript that limits the value to the scientific community.
Major Comments:
The nucleotide sequences used in the analyses are essential for other researchers to replicate or adequately evaluate the work. The authors note that the GenBank accession numbers are pending - they are required for work like this.
The phylogenetic analysis of the NA gene differs substantially from the phylogenetic analyses of the other genes. H5N6 strains from the North-American lineage are included in the NA analysis but not in any other analyses. Conversely, strains such as the A/muscovey duck/Vietnam/LBM635/2014 are included in other analyses but not the NA anaylsis. This makes it impossible for the reader to compare the relationship of the NA genes from the viruses characterized in this work with other viruses. It would be easier to follow if all the phylogenetic trees used the same strains.
Page 13, line 208: The primary citation (Matasuoka Y et al., 2009) should be used when referring to the pathogenicity of strains with the deletion in the NA gene.
Minor comments:
page 2, line 62: the statement that several subtypes (H5N1, H5N2, H5N3...) should be supported by a reference such as the WHO mediacenter/releases/2014/20140507/en and OIE 2015 update on avian influenza.
page 2, line 92: 104 50% egg infectious dose should be 104 50%
page 4, line 151: it is not clear what the 97.5% and 99.9% homology means, perhaps it should read identity and homology respectively.
page 13, line 241 on: In the text and in Tables 1 and 2, please write log10 rather than the short-hand lg, as lg is used for log2 in some countries and log10 in others, and this can create confusion.
The authors highlight genetic differences between all the strains in the text. It would be appropriate for the authors to highlight any genetic differences between GS144 and GS148 that might contribute pathogenicity and transmissibility if there are any.
Author Response
Response to Reviewer 2 Comments
Point 1: The nucleotide sequences used in the analyses are essential for other researchers to replicate or adequately evaluate the work. The authors note that the GenBank accession numbers are pending - they are required for work like this.
Response 1: Thank you for your review. We have uploaded the nucleotide sequences to GenBank. As soon as we get accession numbers, it will be updated immediately.
Point 2: The phylogenetic analysis of the NA gene differs substantially from the phylogenetic analyses of the other genes. H5N6 strains from the North-American lineage are included in the NA analysis but not in any other analyses. Conversely, strains such as the A/muscovy duck/Vietnam/LBM635/2014 are included in other analyses but not the NA analysis. This makes it impossible for the reader to compare the relationship of the NA genes from the viruses characterized in this work with other viruses. It would be easier to follow if all the phylogenetic trees used the same strains.
Response 2: Thank you for your review and your professional suggestion. According to your opinions,All, the phylogenetic trees have been rebuilt as your opinions. we added two new viruses A/mallard/California/1418/2013 (H5N6) and A/mallard/California/1500P/2013(H5N6)
, and they were classified into the North American lineage. The NA gene of A/Muscovy duck/Vietnam/LBM635/2014(H5N1) is N1 subtype, but we added four new N6 subtype viruses(A/Muscovy duck/Vietnam /LBM754 /2014(H5N6), A/Muscovy duck/Vietnam/ LBM755/2014(H5N6), A/Muscovy duck/Vietnam/ LBM756/2014 (H5N6), A/Muscovy duck/ Vietnam/ LBM757/2014 (H5N6)), and they and other viruses, which were not belong to theNorth American lineage, were classified into the Eurasian lineage. We have modified all figures as your opinions in Figure 1-8 of the revised manuscript.
Point 3: Page 13, line 208: The primary citation (Matasuoka Y et al., 2009) should be used when referring to the pathogenicity of strains with the deletion in the NA gene.
Response 3: Thank you for your review. We have corrected the citation as your opinions in the line 226-228 and 380-381of the revised manuscript.
Point 4: page 2, line 62: the statement that several subtypes (H5N1, H5N2, H5N3...) should be supported by a reference such as the WHO mediacenter/releases/2014/20140507/en and OIE 2015 update on avian influenza.
Response 4: Thank you for your review. We have corrected the citation as your opinions in the line 64 of the revised manuscript.
Point 5: page 2, line 92: 104 50% egg infectious dose should be 10450%
Response 5: Thank you for your review. We have corrected the descriptions as your opinions in the line 92 of the revised manuscript.
Point 6: page 4, line 151: it is not clear what the 97.5% and 99.9% homology mean, perhaps it should read identity and homology respectively.
Response 6: Thank you for your review. We have corrected the descriptions as your opinions in the line 154-155 of the revised manuscript.
Point 7: page 13, line 241 on: In the text and in Tables 1 and 2, please write log10rather than the short-hand lg, as lg is used for log2in some countries and log10in others, and this can create confusion.
Response 7: Thank you for your review. We have corrected the descriptions as your opinions in text, Tables 1 and 2 of the revised manuscript.
Point 8: The authors highlight genetic differences between all the strains in the text. It would be appropriate for the authors to highlight any genetic differences between GS144 and GS148 that might contribute pathogenicity and transmissibility if there are any.
Response 8: Thank you for your review. According to your and reviewer 2's opinion, we added the discussion about pathogenicity differences between GS144 and GS148 virus in the line 404-411 of the revised manuscript. We also added the four new Tables (S2-S5) about the amino acids difference between GS144 and GS148 viruses in the supplementary information. Moreover, we have added the statistical analyses in the table 1 of the revised manuscript.

Reviewer 3 Report
In the present manuscript, the authors sequenced the eight segments of 10 H5N6 avian influenza viruses, which were isolated from geese in Guangdong Provinces in 2015 – 2016. The sequences were compiled and edited. They performed the phylogenetic analysis using the distance-based neighbor-joining method and the reliability of the generated trees was assessed by bootstrap analysis using 1000 replicates. Phylogenic analysis showed that HA sequence belonged to the clade 2.3.4.4; The NA and PB2 genes derived respectively from the Eurasian lineage and clade Mix-like 2 (CH.VN.JP), whereas the other segments derived from clade Mix-like 1 (CH.VN.LS). The authors have found the all analyzed isolates displayed a nucleotide sequence homology between 98.0% and 99.9%, making the authors to assume that those viruses probably shared the same ancestor and they resulted from the recombination among different genotypes.
In addition, the authors evaluated if the sequence of H5N6 isolates have some genetic hallmarks, which have been linked to increased virulence, replication in mammals and drug resistance. They concluded, for example, that HA of all isolates harbored a multibasic cleavage site, which is a major feature of highly virulent avian influenza viruses.
The authors chose two H5N6 isolates (named GS144 and GS148) to be evaluated for pathogenicity in chickens. They showed that GS144 was far more virulent than GS148, as evidenced by increased mortality, viral titers in organs and tissues and viral shedding.
The manuscript is interesting. However, some modifications should be done before it is prone to be published.
Major comments:
1)It seems that isolates GS144 and GS148 were chosen because they were completely sequenced. It is not clear if the other isolates were only partially sequenced. If so, the authors should include a table, showing the region of each segment of each isolate that were sequenced. 2)GS144 and GS148 isolates displayed remarkably different pathogenicity in chickens. However, no differences about the sequences of those isolates were highlighted by the authors. Therefore, the authors should include a new table, showing the amino acids difference between the two isolates. Moreover, the authors should discuss how those amino acids could be linked do the increased virulence of GS144 isolate when compared to GS148.
Minor comments:
1)Page 2, lane 92: The number 4 of viral inoculum should be written as superscript.
2)Page 2, lane 92: The authors should specify in which vehicle (buffered saline) the virus has been diluted to be inoculated into the chicken eggs.Author Response
Response to Reviewer 3 Comments
Point 1: It seems that isolates GS144 and GS148 were chosen because they were completely sequenced. It is not clear if the other isolates were only partially sequenced. If so, the authors should include a table, showing the region of each segment of each isolate that were sequenced.
Response 1: Thank you for your review and your professional suggestion. We have determined the whole-genome sequences of our 10 viruses. Our data showed that the 8 segments of our 10 viruses included the nucleotides 1704 nt of HA, 1431 nt of NA, 2280 nt of PB2, 2274 nt of PB1, 2151 nt of PA, 1497 nt of NP, 982 nt of M and 823 nt of NS.
We also added a new Table S1 showed that the 8 segments of our 10 viruses were sequenced in the Supplementary information.
Point 2: GS144 and GS148 isolates displayed remarkably different pathogenicity in chickens. However, no differences about the sequences of those isolates were highlighted by the authors. Therefore, the authors should include a new table, showing the amino acids difference between the two isolates. Moreover, the authors should discuss how those amino acids could be linked do the increased virulence of GS144 isolate when compared to GS148.
Response 2: Thank you for your review and your professional suggestion. We added the four Tables (S2-S5) showing the amino acids different between GS144 and GS148 viruses in the supplementary information.
Point 3: Page 2, lane 92: The number 4 of viral inoculum should be written as superscript.
Response 3: Thank you for your review. We have corrected the descriptions as your opinions in the line 92 of the revised manuscript.
Point 4: Page 2, lane 92: The authors should specify in which vehicle (buffered saline) the virus has been diluted to be inoculated into the chicken eggs.
Response 4: Thank you for your review. We have corrected the descriptions as your opinions in the line 90 of the revised manuscript.

Round 2
Reviewer 1 Report
The authors have not adequately addressed all the points that were raised
More specifically, the remarks are as follows.
Major remarks
Lines 24 and 26. There is a contradiction between the two sentences. The sentence “The two viruses…” suggests that the two viruses have similar transmissibilities, while the sentence “The transmissibility…” states the contrary.Table 1. What exactly is the meaning of EID50/0.1 ml of tissue? It would be more correct to express the viral loads as log10 of EID50/0.1 g of tissue (or log10 of EID50/ g or mg of tissue).Table 2. Again, the authors should try to convert the virus titers in swabs as log10 of EID50/swab.Table S3. What is residue 246 in NS? NS1 from these viruses is a 225 amino-acid protein.Tables S4 and S5 are not at all understandable. What exactly do the substitutions represent? Are they substitutions between A/Gs/Gd/1/1996 and A/Ck/Gd/S1311/2010? And if so, why? The authors should combine the data in tables S4 and S5 in a single table that clearly shows the differences that distinguish GS144 from GS148. Even better, table S3 should be completed with the data from tables S4 and S5.The first 33 lines of the Discussion are redundant with the introduction.Lines 34-60 of the discussion are a mere repetition of the results, with no real discussion.Only lines 76-104 of the discussion are dedicated to the some discussion of the results from the chicken experiment. However, the authors do not discuss in depth their data, and there are several points that need to be discussed in depth:1- The authors should propose hypotheses to try to explain how easy it seems to isolate variants with low pathogenicity phenotypes among a set of only ten HPAIVs.2- There are differences between the two viruses as regards viral shedding in both infected and contact chickens, but the authors overlook these differences (lines 98-99 of the discussion).3- The authors do not even try to propose hypotheses to explain the difference of pathogenicity between the two viruses. At least the comparison of the two full genomic sequences of GS144 and GS148 could point to candidate substitutions associated with the distinct levels of pathogenicity. They should really discuss the data in table S3 to try to point to the substitutions that are more likely to explain the different phenotypes.Minor remarks
Line 22 …intranasally to assess their…Line 24 .. The two viruses were efficiently transmitted to contact chickens.Line 38. http://www.oie.int/en/animal-health-in-the-world/avian-influenza-portal/ and https://www.who.int/influenza/human_animal_interface/en/Line 71. “mortality rate of 37.5%” is redundant with 6 deaths/16 infections.Line 114. ..intranasally .Lines 145-148. Phylogenetic analyses used the nucleotides sequences of the open reading frames (ORFs) of HA (1704 nt), NA (1431 nt), PB2 (2280 nt), …. and of the combined, overlapping ORFs of M1 and M2 (982 nt) and NS1 and NEP (823 nt).Lines 149-171. There are a lot of redundancies in the text. All these informations should be condensed in a table, and commented by a few lines of text.Legends to Figs 1-8. The authors should indicate “The scale bar indicates the branch length and corresponds to 0.005 estimated amino acid substitutions per site.”Line 225. .. glycosylation site in HA1 was found in viruses 16018 and 16120. But which are these viruses? Perhaps GS018 and GS120?Line 230. Remove “which may not increase.. . . .mammals.”Line 247…intranasally with 10^4 EID50 of GS144 and GS148.Line 262. ..the mean viral loads in tissues (expressed as log10 of EID50) were 8.25 in the lungs…Tables 1 and 2. Replace “Contacted” by “Contact”.
Line 4. ..and died in eight days (mortality 87.5%, mean death time 5.5 days).Line 11….contact group, the mean viral loads at five DPI (as expressed in log10 of EID50)..Lines 11-15 of this paragraph are redundant with table 1. This should be shortened and replaced by a synthetic comment to compare the two viruses.Lines 19-31 of this paragraph. Again, this is redundant with table 2. The authors should instead write a synthetic comment to compare the two viruses.Table S1. All the lines are 100% identical. One line is sufficient (and it is redundant with lines 145-48 of the main text).Table S3. H5-numbering (not H5-membering).
Author Response
Response to Reviewer 1 Comments
Major remarks
Point 1: Lines 24 and 26. There is a contradiction between the two sentences. The sentence “The two viruses…” suggests that the two viruses have similar transmissibilities, while the sentence “The transmissibility…” states the contrary.
Response 1: Thank you for your review. We have corrected the descriptions as your opinions in the line 24-27 of the revised manuscript
Point 2: Table 1. What exactly is the meaning of EID50/0.1ml of tissue? It would be more correct to express the viral loads as log10 of EID50/0.1g of tissue (or log10 of EID50/g or mg of tissue).
Response 2: Thank you for your review. We have corrected the descriptions as your opinions in the table 1 of the revised manuscript. For the meaning of EID50/0.1 ml: 1 g of tissue were homogenized and suspended in 1 ml pre-cool isolation media PBS (pH 7.4). and took 0.1 mL from the tissue homogenate to test the viral EID50/0.1 ml.
Point 3: Table 2. Again, the authors should try to convert the virus titers in swabs as log10 of EID50/swab.
Response 3: Thank you for your review. According to the general, most of the studies select the log10EID50/0.1 ml as a unit of viral loads. In our experiment, Oropharyngeal and cloacal swabs were collected from all chickens, and suspended with 1 ml PBS for the detection of viruses shedding. Took 0.1 mL from the suspensions and inoculated into 9 - 11-day-old embryonic eggs incubated at 37°C for 48 hours to test the viral EID50. The log10EID50/0.1 ml it is more appropriate.
Here are the references.
Li, Z., Jiang, Y., Jiao, P., Wang, A., Zhao, F., Tian, G., ... & Chen, H. (2006). The NS1 gene contributes to the virulence of H5N1 avian influenza viruses. Journal of virology, 80(22), 11115-11123.
Chen, H., Deng, G., Li, Z., Tian, G., Li, Y., Jiao, P., ... & Yu, K. (2004). The evolution of H5N1 influenza viruses in ducks in southern China. Proceedings of the National Academy of Sciences, 101(28), 10452-10457.
Jiao, P., Tian, G., Li, Y., Deng, G., Jiang, Y., Liu, C., ... & Chen, H. (2008). A single-amino-acid substitution in the NS1 protein changes the pathogenicity of H5N1 avian influenza viruses in mice. Journal of virology, 82(3), 1146-1154.
Brown, J. D., Stallknecht, D. E., & Swayne, D. E. (2008). Experimental infection of swans and geese with highly pathogenic avian influenza virus (H5N1) of Asian lineage. Emerging Infectious Diseases, 14(1), 136.
Point 4:Table S3. What is residue 246 in NS? NS1 from these viruses is a 225 amino-acid protein.
Response 4:Thank you for your review. We are sorry that we make a mistake. We have corrected the descriptions as your opinions in the new Table S2 of the supplementary information.
Point 5:Tables S4 and S5 are not at all understandable. What exactly do the substitutions represent? Are they substitutions between A/Gs/Gd/1/1996 and A/Ck/Gd/S1311/2010? And if so, why? The authors should combine the data in tables S4 and S5 in a single table that clearly shows the differences that distinguish GS144 from GS148. Even better, table S3 should be completed with the data from tables S4 and S5.
Response 5:Thank you for your review. We added the new Table S3 to showed the summary of signature amino acid Mutation in GS144 and GS148 viruses of the supplementary information.
Point 6: The first 33 lines of the Discussion are redundant with the introduction.
Response 6: Thank you for your review. We have corrected the descriptions as your opinions in the line 328-342 of the revised manuscript
Point 7: Lines 34-60 of the discussion are a mere repetition of the results, with no real discussion. Only lines 76-104 of the discussion are dedicated to some discussion of the results from the chicken experiment. However, the authors do not discuss in depth their data, and there are several points that need to be discussed in depth:
1. The authors should propose hypotheses to try to explain how easy it seems to isolate variants with low pathogenicity phenotypes among a set of only ten HPAIVs.
2. There are differences between the two viruses as regards viral shedding in both infected and contact chickens, but the authors overlook these differences (lines 98-99 of the discussion).
3. The authors do not even try to propose hypotheses to explain the difference of pathogenicity between the two viruses. At least the comparison of the two full genomic sequences of GS144 and GS148 could point to candidate substitutions associated with the distinct levels of pathogenicity. They should really discuss the data in table S3 to try to point to the substitutions that are more likely to explain the different phenotypes.
Response 7: Thank you for your review. For better understand in pathogenicity and transmissibility of the two viruses, we recommend that the description in the discussion line 34-60 and 76-104 may be better.
We have corrected the descriptions as your opinions in the line 409-417 of the revised manuscript.
The idea is excellent. However, in our experiment, our goal was to found the different pathogenicity between GS144 and GS148 with animal experiment, and those data were used to analyze the differences between the two viruses in pathogenicity and transmissibility. We can only discover the differences by experimental results that try to explain it.
Here are the references.
Li, Z., Jiang, Y., Jiao, P., Wang, A., Zhao, F., Tian, G., ... & Chen, H. (2006). The NS1 gene contributes to the virulence of H5N1 avian influenza viruses. Journal of virology, 80(22), 11115-11123.
Chen, H., Deng, G., Li, Z., Tian, G., Li, Y., Jiao, P., ... & Yu, K. (2004). The evolution of H5N1 influenza viruses in ducks in southern China. Proceedings of the National Academy of Sciences, 101(28), 10452-10457.
Jiao, P., Tian, G., Li, Y., Deng, G., Jiang, Y., Liu, C., ... & Chen, H. (2008). A single-amino-acid substitution in the NS1 protein changes the pathogenicity of H5N1 avian influenza viruses in mice. Journal of virology, 82(3), 1146-1154.
Jiao, P., Cui, J., Song, Y., Song, H., Zhao, Z., Wu, S., ... & Liao, M. (2016). New reassortant H5N6 highly pathogenic avian influenza viruses in Southern China, 2014. Frontiers in microbiology, 7, 754.
Brown, J. D., Stallknecht, D. E., & Swayne, D. E. (2008). Experimental infection of swans and geese with highly pathogenic avian influenza virus (H5N1) of Asian lineage. Emerging Infectious Diseases, 14(1), 136.
Li, S., Liu, C., Klimov, A., Subbarao, K., Perdue, M. L., Mo, D., ... & Bryant, M. (1999). Recombinant influenza A virus vaccines for the pathogenic human A/Hong Kong/97 (H5N1) viruses. The Journal of infectious diseases, 179(5), 1132-1138.
Minor remarks
Point 8: Line 22 …intranasally to assess their
Response 8:Thank you for your review. We have corrected the descriptions as your opinions in the line 22 of the revised manuscript
Point 9: Line 24 ...The two viruses were efficiently transmitted to contact chickens
Response 9:Thank you for your review. We have corrected the descriptions as your opinions in the line 24-25 of the revised manuscript
Point 10: Line 38. http://www.oie.int/en/animal-health-in-the-world/avian-influenza-portal/ and https://www.who.int/influenza/human_animal_interface/en/
Response 10:Thank you for your review. We have corrected the descriptions as your opinions in the line 39-40 of the revised manuscript
Point 11: Line 71. “mortality rate of 37.5%” is redundant with 6 deaths/16 infections.
Response 11:Thank you for your review. We have corrected the descriptions as your opinions in the line 72-73 of the revised manuscript
Point 12: Line 114. ...intranasally.
Response 12:Thank you for your review. We have corrected the descriptions as your opinions in the line 115 of the revised manuscript
Point 13: Lines 145-148. Phylogenetic analyses used the nucleotides sequences of the open reading frames (ORFs) of HA (1704 nt), NA (1431 nt), PB2 (2280 nt), …. and of the combined, overlapping ORFs of M1 and M2 (982 nt) and NS1 and NEP (823 nt).
Response 13:Thank you for your review. We have corrected the descriptions as your opinions in the line 147-149 of the revised manuscript.
Point 14: Lines 149-171. There are a lot of redundancies in the text. All these informations should be condensed in a table, and commented by a few lines of text.
Response 14: Thank you for your review. We have corrected the descriptions as your opinions in the line 150-166 of the revised manuscript.
Point 15: Legends to Figs 1-8. The authors should indicate “The scale bar indicates the branch length and corresponds to 0.005 estimated amino acid substitutions per site.”
Response 15:Thank you for your review. We have added the information in the figure 1 to 8of the revised manuscript.
Point 16: Line 225. ... glycosylation site in HA1 was found in viruses 16018 and 16120. But which are these viruses? Perhaps GS018 and GS120?
Response 16:Thank you for your review. We have corrected the descriptions as your opinions in the line 228 of the revised manuscript.
Point 17: Line 230. Remove “which may not increase... mammals.”
Response 17:Thank you for your review. We have corrected the descriptions as your opinions in the line 233 of the revised manuscript.
Point 18: Line 247…intranasally with 10^4 EID50 of GS144 and GS148.
Response 18:Thank you for your review. We have corrected the descriptions as your opinions in the line 249 of the revised manuscript.
Point 19: Line 262. …the mean viral loads in tissues (expressed as log10 of EID50) were 8.25 in the lungs…Tables 1 and 2. Replace “Contacted” by “Contact”.
Response 19:Thank you for your review. We have corrected the descriptions as your opinions in the line 263-264, line 266 and tables 1 and 2 of the revised manuscript.
Point 20:Line 307. ...and died in eight days (mortality 87.5%, mean death time 5.5 days).
Response 20:Thank you for your review. We have corrected the descriptions as your opinions in the line 302 of the revised manuscript.
Point 21:Line 315… contact group, the mean viral loads at five DPI (as expressed in log10 of EID50) ...
Response 21:Thank you for your review. We have corrected the descriptions as your opinions in the line 308 of the revised manuscript.
Point 22: Lines 312-322 of this paragraph are redundant with table 1. This should be shortened and replaced by a synthetic comment to compare the two viruses.
Response 22: Thank you for your review. We have corrected the descriptions as your opinions in the line 306-313 of the revised manuscript.
Point 23:Lines 323-328 of this paragraph. Again, this is redundant with table 2. The authors should instead write a synthetic comment to compare the two viruses.
Response 23: Thank you for your review. For better understand the shedding ability of the two viruses, we recommend that the description in the line 314-326 may be better.
Point 24: Table S1. All the lines are 100% identical. One line is sufficient (and it is redundant with lines 145-148 of the main text).
Response 24: Thank you for your review. We have corrected the descriptions as your opinions in the line 147-149 of the revised manuscript. We have deleted the descriptions as your opinions intheTable S1of the revised manuscript.
Point 25: Table S3. H5-numbering (not H5-membering).
Response 25:Thank you for your review. We have corrected the descriptions as your opinions in the new Tables S2 of the revised manuscript.
Point 26: Extensive editing of English language and style required
Response 26: Thank you for your review. Our manuscript has been re-edited for English language in the https://americanmanuscripteditors.com/.

Reviewer 2 Report
Thank you to the authors for addressing the reviewer concerns, including the addition of the supplementary tables. The manuscript is improved although there are still some minor edits.
line 90 and 120: should be "pre-cooled"
figure 1 has errors with the naming of A/goose/Guangdong/GS114 (2015 vs 2016) and is missing A/goose/Guangdong/GS144/2015
line 225: viruses 16018 and 16120 should be named GS018 and GS120?
If possible, the column width should be adjusted in tables 1 and 2 so that there are no isolated parentheses )
Page 17: "seven of the eight non-contact chickens" rather than 'eighth not'
Page 19: remove 'And' from the beginning of the sentence "The mortality rate of GS144..."
Author Response
Response to Reviewer 2 Comments
Point 1: line 90 and 120: should be "pre-cooled"
Response 1: Thank you for your review. We have corrected the descriptions as your opinions in the line 92 and line 122 of the revised manuscript.
Point 2: figure 1 has errors with the naming of A/goose/Guangdong/GS114 (2015 vs 2016) and is missing A/goose/Guangdong/GS144/2015
Response 2:Thank you for your review. We have corrected the descriptions as your opinions in the figure 1 of the revised manuscript.
Point 3: line 225: viruses 16018 and 16120 should be named GS018 and GS120?
Response 3: Thank you for your review. We have corrected the descriptions as your opinions in the line 228 of the revised manuscript.
Point 4: If possible, the column width should be adjusted in tables 1 and 2 so that there are no isolated parentheses)
Response 4: Thank you for your review. We have corrected the descriptions as your opinions in the table 1 and 2 of the revised manuscript.
Point 5: Page 17: "seven of the eight non-contact chickens" rather than 'eighth not'
Response 5: Thank you for your review. We have corrected the descriptions as your opinions in the line 300-301 of the revised manuscript.
Point 6:Page 19: remove 'And' from the beginning of the sentence "The mortality rate of GS144..."
Response 6: Thank you for your review. We have corrected the descriptions as your opinions in the line 386-387 of the revised manuscript.
